# The Bactericidal Effect of a Combination of Food-Grade Compounds and their Application as Alternative Antibacterial Agents for Food Contact Surfaces

**DOI:** 10.3390/foods9010059

**Published:** 2020-01-07

**Authors:** Kyung Min Park, Sung-Geon Yoon, Tae-Ho Choi, Hyun Jung Kim, Kee Jai Park, Minseon Koo

**Affiliations:** 1Department of Food Biotechnology, University of Science & Technology, Daejoen 34113, Korea; kyungmuni82@naver.com (K.M.P.); hjkim@kfri.re.kr (H.J.K.); 2Department of Research Group of Consumer Safety, Korea Food Research Institute, Wanjugun 55365, Korea; jake@kfri.re.kr; 3Antibacterial Materials Lab., DYNE SOZE Co., Ltd., R&D Center, Yongin 16827, Korea; 1226pkm@hanmail.net (S.-G.Y.); thchoi@dynesoze.co.kr (T.-H.C.); 4Department of Food Analysis Center, Korea Food Research Institute, Wanjugun 55365, Korea

**Keywords:** food-grade antibacterials, bactericidal effect, food contact surface, foodborne pathogen, chemical sanitizers

## Abstract

Chemical antibacterials are widely used to control microbial growth but have raised concerns about health risks. It is necessary to find alternative, non-toxic antibacterial agents for the inhibition of pathogens in foods or food contact surfaces. To develop a non-toxic and “green” food-grade alternative to chemical sanitizers, we formulated a multicomponent antibacterial mixture containing *Rosmarinus officinalis* L., *Camellia sinensis* L., citric acid, and ε-polylysine and evaluated its bactericidal efficacy against *Staphylococcus aureus*, *Escherichia coli*, *Bacillus cereus*, *Salmonella* Enteritidis, and *Listeria monocytogenes* on food contact surfaces. A combination of the agents allowed their use at levels lower than were effective when tested individually. At a concentration of 0.25%, the multicomponent mixture reduced viable cell count by more than 5 log CFU/area, with complete inactivation 24 h after treatment. The inhibitory efficacy of the chemical antibacterial agent (sodium hypochlorite, 200 ppm) and the multicomponent antibacterial mixture (0.25%) on utensil surfaces against *S. aureus*, *E. coli*, *S.* Enteritidis, and *L. monocytogenes* were similar, but the multicomponent system was more effective against *B. cereus* than sodium hypochlorite, with an immediate 99.999% reduction on knife and plastic basket surfaces, respectively, and within 2 h on cutting board surfaces after treatment. A combination of these food-grade antibacterials could be a useful strategy for inhibition of bacteria on food contact surfaces while allowing use of lower concentrations of its components than are effective individually. This multicomponent food-grade antibacterial mixture may be a suitable “green” alternative to chemical sanitizers.

## 1. Introduction

Foodborne diseases caused by pathogens present in foods are considered an emerging public health problem and encompass a wide spectrum of diseases [1]. These diseases result from the consumption of food contaminated with foodborne pathogens or toxic compounds derived from these pathogens. These diseases are often spread by inadequate hygiene practices and bacterial cross-contamination between foods and food surface materials in the food-processing industry [2]. Food contact surfaces can provide a suitable substrate for bacterial attachment and are potential sources of cross-contamination, which is a significant concern for the food industry [3,4].

Many manufacturing facilities include materials such as stainless steel (SS) (e.g., washing tanks), rubber surfaces (e.g., belts, packing machines and liners), and utensils [5], which provide niches for bacterial cross-contamination between food and food contact surfaces. Although European and US legislations have strict regulations concerning materials that come in contact with food, epidemic outbreaks caused by foodborne pathogens are frequently reported [6,7]. Several studies have found that *Escherichia coli*, *Staphylococcus aureus*, *Salmonella* spp., and *Listeria monocytogenes* survive on hands, cloths, utensils, and food-processing facility surfaces for hours or days after initial contact with the microorganisms. To prevent food poisoning outbreaks caused by foodborne pathogens, it is important to inhibit bacterial attachment to food contact surfaces and bacterial contamination of food sources [8]. The prevention of bacterial attachment and contamination can be facilitated by the regular sanitization and disinfection of surfaces [9].

In the food industry, chemical antibacterials, such as ethyl alcohol, organic acids, hydrogen peroxide, and quaternary ammonium, are routinely used to eliminate foodborne pathogens [10]. However, ethyl alcohol and isopropyl alcohol, as disinfectants, exhibit slow action against nonenveloped viruses, a lack of sporicidal activity, reduced activity in the presence of organic matter, and adverse effects on some types of medical equipment [11]. Chlorine has been associated with the formation of toxic compounds, such as trihalomethanes [12], and organic acids may cause equipment damage and represent toxic hazards to workers. The World Health Organization (WHO) is promoting the development of alternative methods for the bacterial inhibition of food or food contact surfaces [13]. Food-grade antibacterial materials, such as plant extracts, organic acid, and food additives have been increasingly reported as effective alternatives to current chemical antibacterials due to their safety and nontoxic status.

Green tea (*Camellia sinensis* L.) has also been reported to exhibit antibacterial activity against various pathogenic bacteria via the action of catechins (epicatechin-3-gallate, ECG; epigallocatechin, EGC; EGC-3-gallate, EGCG) [14]. The extracts of *C. sinensis* L. have been shown to inhibit the growth of *S. aureus*, *E. coli*, *Bacillus subtilis*, and *Bacillus cereus* in many studies [15,16,17,18]. *Rosmarinus officinalis* L. is well known for its antibacterial properties and has been widely commercialized as a preservation agent in the food industry [19,20]. *R. officinalis* itself contains a variety of polyphenolic compounds, such as carnosic acid and carnosol [21]. Plant-derived compounds exhibit significant antibacterial activity against both Gram-positive and Gram-negative bacteria [22,23]. The extracts of *R. officinalis* L. and *C. sinensis* L. have been approved by the European Union (EU) for use in food preservation and have been categorized by the FDA as Generally Recognized as Safe (GRAS). Organic acids are natural food additives and are generally considered GRAS by the United States Food and Drug Administration [24]. As a weak organic acid, citric acid can cross the cell membrane and lower intracellular pH. A low pH within cells causes damage to DNA, protein, and extracellular membranes, leading to the death of bacteria, such as *E. coli* O157: H7, *Salmonella enterica* subsp. *enterica* serovar Typhimurium, and *Staphylococcus aureus* [20,25,26]. ε-polylysine is a natural antimicrobial cationic peptide which is generally regarded as safe as a food preservative. ε-polylysine can inhibit various microorganisms, including bacteria, yeasts, and viruses. The antimicrobial effect of ε-polylysine has been associated with its cationic charge, as this allows it to adsorb onto negatively charged microbial surfaces and disrupt the cell membrane, envelope or cell walls. [27,28,29].

When antibacterial agents are combined, different interactions may occur, with various effects that may be synergistic, antagonistic, or additive [30]. The combination of antibacterial agents shows more powerful effects that enhance antibacterial activity, allowing the use of lower doses of each plant extract and food-grade antibacterial agent. Despite the valuable information obtained on the antibacterial activity of *R. officinalis* L., *C. sinensis* L., citric acid, and ε-polylysine, the combination of food-grade antibacterials has not yet been studied.

The aim of the present study was to evaluate the antibacterial properties of the multicomponent food-grade antibacterial mixture that exhibits a greater bactericidal effect than those of the individual materials used alone. In addition, this study evaluated the potential of the formulated antibacterial mixture as an alternative approach to inactivating bacteria on food contact surfaces.

## 2. Materials and Methods

### 2.1. Preparation of Food-Grade Antibacterials

*R. officinalis* L. and *C. sinensis* L. extracts were obtained from Dyne Soze (Yongin, Korea). *R. officinalis* L. was extracted with ethanol and contained 20% carnosic acid. *C. sinensis* L. was extracted with methanol. Citric acid was obtained from Sigma–Aldrich (St Louis, MO, USA). The extracts of *R. officinalis* L. and *C. sinensis* L. were dissolved in ethanol respectively, as solvents to make 10% (*w*/*v*) stock solutions. The citric acid and ε-polylysine were dissolved in distilled water (DW) to prepare a 10% (*w*/*v*) stock solution. All disinfectant solutions were filtered with a maximum pore diameter of 0.45 µm (Millipore Co., Billerica, MA, USA).

### 2.2. Formulation of Multicomponent Antibacterial Mixture

The nonalcoholic and multicomponent antibacterial mixture mixture was composed of *R. officinalis* L. extract, *C. sinensis* L. extract, citric acid, and ε-polylysine. An additional compound, glycerin fatty acid ester, was used to assist in the emulsification of the formulations. A 50X stock solution was prepared from 0.25% (*w*/*v*) for *R. officinalis* L., 0.5% (*w*/*v*) for *C. sinensis* L., 20% (*w*/*v*) for citric acid, 0.5% (*w*/*v*) for ε-polylysine, and 25% (*w*/*v*) for glycerin fatty acid ester with DW. The formulation process for this multicomponent antibacterial mixture was patented in 2018 [31].

### 2.3. Bactericidal Activity of Food-Grade Antibacterials

#### 2.3.1. Bacterial Strains

Five strains were used to evaluate the antibacterial effects of individual disinfectant agents: *E. coli* ATCC 10536, *B. cereus* ATCC 14579, *L. monocytogenes* ATCC 15313, *S. aureus* ATCC 6538, and *Salmonella enterica* subsp. *enterica* serovar Enteritidis ATCC 13076. All microbial strains were subcultured in tryptic soy broth (TSB, Merck, Darmstadt, Germany) and incubated at 37 °C for *S*. Enteritidis, *E. coli*, and *S. aureus* and at 30 °C for *B. cereus* and *L. monocytogenes.* The cultures were washed twice in sterile isotonic saline solution (0.85% NaCl) by centrifugation at 1570× *g* for 20 min, and the pellets were resuspended in 1 mL of sterile phosphate-buffered saline (PBS) at approximately 8–9 log CFU/mL.

#### 2.3.2. Bactericidal Activity of Food-Grade Antibacterials

The antibacterial activity testing procedure was modified from Prabuseenivasan et al. [32]. Briefly, the bacterial suspension was adjusted to McFarland standard 0.5 (approximately 10^6^ CFU/mL) and spread over Muller Hinton Agar (MHA; Merck, Darmstadt, Germany) plates using a sterile cotton swab. Each antibacterial material (*R. officinalis* L., *C. sinensis* L. and citric acid) was prepared at a 0.5% (*w*/*v*) concentration and sterilized by filtration. Sterilized disks (Whatman No. 5, 6 mm diameter) were impregnated with 20 µL of antibacterial materials and placed on the MHA surface; 10% DMSO used as a negative control. After incubation at 37 °C or 30 °C for 20 h, the inhibition zone was measured. All experiments were performed independently in triplicate, and the mean value was calculated.

#### 2.3.3. Minimum Inhibitory Concentration (MIC) Determination

The antibacterial activity of *R. officinalis* L., *C. sinensis* L., citric acid, and the multicomponent antibacterial mixture against foodborne pathogens was tested by determining the minimum inhibitory concentration (MIC) in 96-well flat-bottom plates (BD Biosciences, Basel, Switzerland) according to the CLSI broth microdilution method [33]. Overnight cultures were prepared by incubating the inocula and adjusting to the 0.5 McFarland standard. In each well of a 96-well plate, 100 µL of TSB was dispensed, followed by the addition of 100 µL of antibacterial material (stock solution) into the first well and serial dilution to achieve the final concentrations ranging from 5% to 0.0192% (*w*/*v*) for *R. officinalis* L., *C. sinensis* L., ε-polylysine, and the multicomponent antibacterial mixture. Citric acid solutions were prepared at concentrations ranging from 8% to 0.03125% (*w*/*v*) for MIC determination. Subsequently, 10 µL of the bacterial suspension (0.5 McFarland) was added to each well, and the microtiter plates were incubated at the optimal temperature for each pathogen for 18–24 h. The last well, as the negative control, contained 200 µL of TSB alone without any antibacterial material. The lowest concentration that completely inhibited visible growth was established as the MIC.

### 2.4. Antibacterial Activity of the Formulated Multicomponent Antibacterial Mixture

#### 2.4.1. Surface Material Preparation

Cutting boards (polypropylene, PP), knives (stainless steel, SS), plastic baskets (PP), SS (ASI type 304), and PP were selected as representative food contact surfaces used in the food industry. The cutting boards were divided into surfaces of specific dimensions (10 × 10 cm, 100 cm^2^) to be used for the experiments. The blades of the knives had surfaces that were 4 cm in width and 7.5 cm in length (i.e., a total surface area of 30 cm^2^), and the plastic baskets were cut to obtain a total surface area of 100 cm^2^ (10 × 10 cm). The SS and PP were cut into coupons with surfaces of 2 × 7 × 0.2 cm. These coupons were then washed with 70% ethyl alcohol for 10 min, followed by two rinses in sterile DW. The cleaned and washed surface materials were air-dried in a laminar flow biosafety cabinet for 2 h and then sterilized by autoclaving (121 °C, 15 min).

#### 2.4.2. Bacterial Survival on Stainless Steel (SS) and Polypropylene (PP)

To assess the antibacterial efficacy of SS and PP, we used modified versions of the techniques described by Kim et al. [34]. Single colonies of *S. aureus, E. coli, B. cereus, S.* Enteritidis, and *L. monocytogenes* were selected and inoculated into 10 mL of sterile TSB. The cultures were incubated at 37 °C for *S.* Enteritidis, *E. coli*, and *S. aureus* and at 30 °C for *B. cereus* and *L. monocytogenes*. A 100 µL suspension was transferred to 10 mL of fresh TSB and incubated in the same manner before bacterial inoculation on the surface. The bacterial suspension was pelleted by centrifugation at 3000× *g* for 20 min at 4 °C, and the pellets were washed twice with sterilized PBS before final resuspension in 10 mL of sterile 0.85% NaCl. The bacterial suspensions were adjusted to a concentration between 6 and 7 log CFU/mL. The prepared SS and PP coupons were placed in Petri dishes, and 100 µL of bacterial suspension (7 log CFU/g) was inoculated on the surface. The coupons were dipped in 12 mL of the multicomponent antibacterial mixture at 0.25%, 0.5%, and 1% (*w*/*v*) and incubated for 5 min. The coupons were then removed from the Petri dish using flame-sterilized forceps and washed in sterile DW to remove the remaining unattached antibacterials on the surface. The coupons were incubated at 37 °C for *S.* Enteritidis, *E. coli*, and *S. aureus* and at 30 °C for *B. cereus* and *L. monocytogenes* for 1, 4, 12, and 24 h. The coupons were then placed in sterile 50 mL conical centrifuge tubes containing 30 mL of PBS and 2 g of sterile glass beads (Sigma–Aldrich, St. Louis, MO, USA) and then agitated for 5 min with a bench-top vortex mixer set at maximum speed to detach the cells from the coupons. The cell suspensions in the tubes were serially diluted tenfold in 0.85% NaCl and examined in duplicate using 3M Petrifilm™ Aerobic Count Plates (3M Petrifilm, St. Paul, MN, USA). The plates were incubated at 37 °C for *S.* Enteritidis, *E. coli*, and *S. aureus* and at 30 °C for *B. cereus* and *L. monocytogenes* for 24–48 h. Then, the mean viable bacterial counts were determined as log CFU/coupon.

#### 2.4.3. Bacterial Survival on Cutting Boards, Knives, and Plastic Baskets

The bacterial suspensions were prepared by the method described in Section 2.3.1 and adjusted to concentrations ranging from 6 to 7 log CFU/mL. Then, 1.0 mL (cutting board and plastic basket) and 0.5 mL (knife) of each bacterial suspension (6–7 log CFU/mL) were used for inoculation, providing a final population of 5–6 log CFU/cm^2^. After 1 h of rest for bacterial attachment, the cutting boards, knives, and plastic baskets were entirely dipped in 200 ppm sodium hypochlorite (10%, Sigma–Aldrich) and 0.25% of the multicomponent antibacterial mixture for 5 min. The viable bacterial counts were determined at 1, 2, 4, 8, and 24 h. Each area was swabbed using separate sterile wet cotton swabs that were previously prepared in 10 mL of sterile D/E Neutralizing Broth (Difco Laboratories, Detroit, MI, USA). The swabs with the broth were vortexed for 5 min, and 1 mL of the homogenized suspension was serially diluted in 9 mL of 0.85% NaCl. Samples were examined in duplicate using a 3M Petrifilm™ Aerobic Count Plate (3M Petrifilm, St. Paul, MN, USA). The plates were then incubated at optimal temperature for 48 h. The mean viable bacterial counts were determined as log CFU/100 cm^2^ for the cutting boards and plastic baskets and log CFU/30 cm^2^ for the knives. Surfaces that were not treated with disinfectant solutions served as the controls. All tests were carried out in duplicate.

### 2.5. Statistical Analysis

All assays were performed in triplicate in two independent experiments, and the results were expressed as the average and log-transformed values. SPSS statistical software was used to evaluate the results by analysis of variance (ANOVA). To compare the means, Duncan’s test was used with a significance level of *p* = 0.05.

## 3. Results and Discussion

### 3.1. Antibacterial Effect of Food-Grade Antibacterial Materials

Food-grade antibacterial materials were investigated to evaluate their antibacterial activity against foodborne pathogens, including three strains of Gram-positive bacteria (*B. cereus*, *S. aureus*, and *L. monocytogenes*) and two strains of Gram-negative bacteria (*E. coli* and *S.* Enteritidis) using the disk diffusion method. This evaluation of antibacterial activity is recorded in Table 1. The results revealed that the tested antibacterial materials were potentially effective at suppressing the growth of foodborne pathogens with variable potency. The extract of *R. officinalis* L. was the most effective material, retarding the growth of both Gram-positive and Gram-negative microbial pathogens at a concentration of 0.5%. The biological activities of rosemary (*R. officinalis* L) extracts are directly related to the presence of carnosic acid as a major phenolic component. Several scientific publications disagree about the relationships that may exist between the composition of polyphenolic extracts and their antibacterial activity. Moreno et al. [34] demonstrated that the antibacterial efficacy of *R. officinalis* L. is related to the synergy between rosmarinic acid, carnosic acid, and carnosol. The extract of *C. sinensis* L. was effective against only *B. cereus* and *S.* Enteritidis. Epigallocatechin gallate (EGCG) is the most common substance in green tea leaves and adds to antibacterial activity. This active compound is capable of inhibiting the growth of *Bacillus*, *Camplylobacter*, *Clostridium*, *E. coli*, *L. monocytogenes*, *S. aureus*, *Salmonella*, and *Vibrio* [15,18]. The antibacterial activity of citric acid has been previously reported [20,25,26]. In this study, citric acid exhibited inhibitory effects against *B. cereus*, *S. aureus*, and *S.* Enteritidis. Citric acid is a weak acid that has been used as an antibacterial food preservative mainly due to its capacity to inhibit bacterial growth through its disruptive effect on cell membranes [1]. The antibacterial activity of ε-polylysine has been demonstrated in previous studies [17,27,28,29]. However, the ε-polylysine that was evaluated in this study at a concentration of 0.5% was not effective against foodborne pathogens. The observed antibacterial activity of the food-grade antibacterial materials suggests that *R. officinalis* L. was the most effective antibacterial material against both Gram-positive and Gram-negative pathogens. Hence, experiments were conducted to determine the MIC of each antibacterial material against foodborne pathogens.

### 3.2. MICs of Food-Grade Antibacterials

The inhibitory effect of *R. officinalis* L. started at 0.04% against *B. cereus*, *S. aureus*, and *L. monocytogenes*, followed by 0.08% against *S.* Enteritidis and 0.16% against *E. coli* (Table 2), while *C. sinensis* L. and citric acid suppressed the growth of these bacterial pathogens at a concentration of approximately 0.3%. These results indicate that Gram-positive bacteria were more susceptible to antibacterial agents than Gram-negative bacteria, which is consistent with the results of other studies [16,35,36,37]. According to Archana and Abraham et al. [16], the MIC values of a methanolic *C. sinensis* L. extract were 0.8 mg/mL against *E. coli*, 0.8 mg/mL against *S. aureus*, and 1.2 mg/mL against *Salmonella* Typhi. An ethanolic extract of *C. sinensis* L. leaves generated a larger inhibition zone against *E. coli* (13 mm) than against *S. aureus* (12 mm) [1]. An alcoholic extract of *R. officinalis* L. showed greater antibacterial activity against Gram-positive bacteria (MIC 0.20–0.48 mg/mL) than against Gram-negative bacteria (MIC 1.16–1.72 mg/mL) [36]. The MIC of *R. officinalis* L. was 5 mg/mL against *L. monocytogenes* and *E. coli* and 10 mg/mL against *S.* Enteritidis.

Generally, different crude extracts varied in their activities against the same tested microbes [35]. These inconsistencies might be due to differences in the antibacterial activity of the polyphenolic compounds present in the extracts [38]. Nevertheless, these results indicated that Gram-positive bacteria, especially *B. cereus*, are more susceptible to antibacterial materials than Gram-negative bacteria. Gram-negative bacteria have a unique outer membrane that can act as a barrier, so they tend to be more resistant to disinfectant agents than Gram-positive bacteria [39,40].

### 3.3. MICs of the Multicomponent Antibacterial Mixture

Plant extracts, including *R. officinalis* L. and *C. sinensis* L. extracts, are good antibacterial agents, but naturally derived agents tend to be more expensive than synthetic disinfectant agents. It can be difficult to attain the necessary levels of naturally derived antibacterial agents without them becoming cost-prohibitive. Thus, we evaluated the lowest concentration for producing a 5 log CFU/mL reduction in the counts of *E. coli* and *S. aureus* in 5 min based on the time-kill rate and formulated multicomponent antibacterial mixture (50X stock solution) using *R. officinalis* L. extract (0.25%, *w*/*v*), *C. sinensis* L. extract (0.5%, *w*/*v*), and citric acid (25%, *w*/*v*) with glycerin fatty acid ester (25%, *w*/*v*) and ε-polylysine (0.5%, *w*/*v*) in a previous study [31].

As shown in Table 3, the MIC value of the multicomponent antibacterial mixture was 1% against *S. aureus* and 0.5% against *L. monocytogenes*, *E. coli*, and *S.* Enteritidis. The MIC against *B. cereus* was the lowest, with a value of 0.25%. According to the results shown in Table 2, *B. cereus* was susceptible to all the antibacterial materials tested, and the multicomponent antibacterial mixture showed high antibacterial efficacy against *B. cereus*. In addition, the multicomponent antibacterial mixture also exhibited effective antibacterial activity against Gram-negative bacteria, such as *E. coli* and *S.* Enteritidis, as well as against Gram-positive bacteria. Gram-negative bacteria are generally less susceptible to antibacterial plant extracts than Gram-positive bacteria, as their outer membrane, consisting of lipoproteins and lipopolysaccharides, acts as a barrier to antibacterial agents [41]. The multicomponent antibacterial mixture had a positive bactericidal effect against Gram-negative as well as Gram-positive bacteria and could help reduce the necessary concentrations of the *R. officinalis* L. extract, *C. sinensis* L. extract, citric acid, and ε-polylysine for antibacterial activity compared with the use of individual antibacterial agents. In some studies, researchers have suggested that various active components in combined antibacterial agents have stronger antibacterial effects than individual components, suggesting that minor components in the plant extracts are also crucial for the observed activity [30]. The inhibitory effect of the *R. officinalis* L. extract was higher at low pH and high NaCl concentrations than under other conditions [20]. Antibacterial agents, such as citric acid, on the outer membrane of Gram-negative bacteria induce changes in the intracellular pH. Low-pH conditions within cells caused by citric acid treatment induce damage to the extracellular membrane and suppress NADH oxidation, eventually leading to cell death [20]. The ε-polylysine can change the cell wall composition of bacteria and make the cell wall more fragile, thus increasing the cell wall’s permeability [27,29]. Plant extracts also induced cell membrane damage, increasing outer and inner membrane permeability and disrupting the cell membranes [28]. Although ε-polylysine alone did not exhibit effective antibacterial activity against foodborne pathogens, a combination of ε-polylysine and plant extracts may damage the bacterial cell membrane through interaction of its active compounds, facilitating the absorption of active compounds into cells and increasing the permeability of the cell membrane. Thus, the inclusion of citric acid and ε-polylysine in antibacterials formulation may be an effective inhibitory strategy against Gram-positive and Gram-negative bacteria and may accentuate the antibacterial effect of *R. officinalis* L. and *C. sinensis* L. extracts.

### 3.4. Evaluation of Microbial Survival on SS and PP Treated with the Multicomponent Antibacterial Mixture

Studies evaluating plant extracts, organic acids, and food additives for bacterial reduction on food contact surfaces remain somewhat limited. Based on the antibacterial effects of the multicomponent antibacterial mixture against foodborne pathogens, the multicomponent antibacterial mixture could potentially be used as alternative antibacterials for food contact surfaces. We performed an antibacterial activity test against foodborne pathogens attached to SS and PP, and different concentrations of the multicomponent antibacterial mixture (1%, 0.5%, and 0.25%) were used to determine the effective concentration to inactivate viable cells (Figure 1). Against all foodborne pathogens attached on the SS surface, the multicomponent antibacterial mixture showed a reduction of more than 5 log CFU/coupon at the tested concentrations. The multicomponent antibacterial mixture at the lowest concentration of 0.25% immediately inhibited the survival of *E. coli* and *L. monocytogenes* after treatment, and the antibacterial activity was maintained for 24 h. The reduction rates for *B. cereus* and *S.* Enteritidis were 73% and 65% after treatment with a concentration of 0.25% but decreased to less than 1 log CFU/coupon within 1 h. Thereafter, these bacteria were not detected on the SS surface at 24 h. The levels of all the bacteria tested decreased to less than 1 log CFU/coupon within 1 h when the concentrations of the multicomponent antibacterial mixture were 0.5% and 1%. On PP surfaces, treatment with 0.25% was effective for inactivation of the tested pathogens, as this concentration completely inactivated the initially inoculated *E. coli*, *B. cereus*, *S.* Enteritidis, and *L. monocytogenes* within 1 h. When a concentration of 0.25% was used, *S. aureus* survived on the PP surface for 24 h, but the level of *S. aureus* rapidly decreased to 1.0 log CFU/coupon from the initial inoculum level of 7.2 log CFU/coupon within 1 h, and this level was maintained at 1 log CFU/coupon for 24 h. The multicomponent antibacterial mixture may have an excellent ability to reduce the surface colonization and cross-contamination of foodstuffs by foodborne pathogens, thereby preventing foodborne illness. A similar observation was made in a previous study regarding the ability of plant extracts to prevent the attachment of *E. coli*, *B. cereus*, and *S. aureus* [42]. The efficacy of plant extracts against microbial attachment was reported by Vazquez-Sanchez et al. [43], who examined the antibiofilm activity of *R. officinalis* L. extract against *S. aureus* on SS surfaces. Another study examined the efficacy of antibacterial formulation based on plant extracts against the attachment of different bacterial species, such as *Staphylococcus simulans*, *Lactobacillus fermentum*, *Pseudomonas putida*, *S. enterica*, and *L. monocytogenes* [44,45].

Bacterial survival after sanitization represents a potential risk to the food industry and the consumer [26]. It must be emphasized that an appropriate and efficient hygiene protocol is of fundamental importance; the American Public Health Association [46] recommends a maximum limit of 2 CFU/cm^2^ for a food contact surface to be deemed appropriate, whereas the WHO suggests a limit of 30 CFU/cm^2^ [13]. Based on the results obtained, the combined mixture with food-grade antibacterial materials used in this study inactivated almost all viable cells of *S. aureus*, *E. coli*, *B. cereus*, *S.* Enteritidis, and *L. monocytogenes* attached onto the SS and PP surfaces. Thus, the present research evaluated the possibility of using the multicomponent antibacterial mixture from this study as an alternative antibacterials for utensils used in the food-processing industry.

### 3.5. Evaluation of Microbial Survival on Cutting Boards, Knives, and Plastic Baskets Treated with the Multicomponent Antibacterial Mixture

Instruments and materials used in the food-processing industry can be vehicles for pathogenic contamination. Our study selected cutting boards, knives, and plastic baskets to evaluate the use of alternative antibacterials for food contact surfaces. As shown in Figure 2, nontreatment (NT) was complementary to adhesion on the tested utensils’ surfaces because pathogens inoculated on the surfaces of cutting boards, knives, and plastic baskets survived for 24 h, except for *S.* Enteritidis on the cutting board and *E. coli* on the knife. Several studies have indicated that *E. coli*, *S. aureus* and *S.* Enteritidis survive on hands, sponges/cloths, utensils, and currency for hours or days after initial contact with these materials [47]. Montville et al. [48] quantified cross-contamination between the hands and foods or various kitchen surfaces and foods. Infection with *Salmonella* or *Campylobacter* in the Netherlands was caused by cross-contamination directly or indirectly from raw poultry via contaminated surfaces or niches in food processing for ready-to-eat products [8]. These results suggest the importance of hygiene procedures for surfaces that come in contact with food because foodborne pathogens can survive for long periods.

The present study compared the antibacterial effects of the multicomponent antibacterial mixture (0.25%) and sodium hypochlorite (200 ppm) against foodborne pathogens attached on food contact surfaces to evaluate the applicability of this combined antibacterial as an alternative to chemical antibacterial agents. Based on the antibacterial activity, Table 4 shows the time (h) required for 99.999% foodborne pathogen reduction (T_99.999%_) by treatment with sodium hypochlorite and the multicomponent antibacterial mixture. Both sodium hypochlorite and the multicomponent antibacterial mixture immediately reduced the initially attached *S. aureus*, *E. coli*, and *L. monocytogenes* counts to below the level of detection (<1 log CFU/area) after treatment (0_T_), and these bacteria were no longer detected after 24 h. The multicomponent antibacterial mixture was more effective against *B. cereus* on all tested food contact surfaces than sodium hypochlorite at 200 ppm. A 99.999% reduction in *B. cereus* levels was observed immediately after treatment on the knife and plastic basket and within 2 h after treatment on the cutting board. The antibacterial efficacies of the chemical and the multicomponent antibacterial mixture on the knife and plastic basket against *S.* Enteritidis were similar (immediate 99.999% reduction after treatment)*,* but the multicomponent antibacterial mixture more effectively inhibited the attachment of *S.* Enteritidis (0_T_) on the cutting board than sodium hypochlorite (1 h). The growth of foodborne pathogens was inhibited after treatment with the multicomponent antibacterial mixture for 24 h. Sodium hypochlorite (NaOCl) is the most widely used disinfectant in the food industry due to its strong oxidizing capacity [49]. NaOCl disrupts the plasma membranes of bacterial cells and disables the enzymatic active site [50]. However, chemical disinfectants based on chlorine, chloramines, and chlorine dioxide produce unwanted disinfection byproducts (DBPs) when reacting with natural organic matter, anthropogenic contaminants, bromide, or iodide present in the source water [12]. These DBPs may themselves be harmful and have carcinogenic, mutagenic, or genotoxic properties [51]. The alternative antibacterial agents studied here exhibit disinfection qualities comparable to those of traditional disinfectants and antibacterials. In the nontreatment group, the pathogens survived on cutting boards, knives, and plastic baskets for 24 h, but the multicomponent antibacterial mixture completely inactivated the pathogens on the food contact surfaces after treatment. In Korea, utensils used during food processing are typically sanitized by immersion (5 min) in a “utensil sterilizer” containing 200 ppm sodium hypochlorite solution for microbial reduction. In the present study, the multicomponent antibacterial mixture effectively inhibited bacterial attachment to the food contact surfaces. To our knowledge, this is the first investigation of the antibacterial effectiveness of a combined mixture using food-grade antibacterial materials against major foodborne pathogens, including spore-forming bacteria, on the surfaces of utensils, such as cutting boards, knives, and plastic baskets. We suggest that the multicomponent antibacterial mixture tested in the present study could be used as an alternative to chemical antibacterials for killing and inhibiting the growth of foodborne bacteria on food contact surfaces.

## 4. Conclusions

The main aim of this study was to assess the antibacterial effects of a green and non-toxic combination of food-grade compounds as a potential alternative to chemical sanitizers for control of foodborne pathogens attached to food contact surfaces. The combination of *R. officinalis* L. extract, *C. sinensis* L. extract, citric acid, and ε-polylysine achieved greater bactericidal effects at a low concentration compared to their individual use. It was further found that the formulated mixture of food-grade compounds may have potential as an alternative to chemical sanitizers. Compared to sodium hypochlorite, which is generally used in the food-processing industry, the multicomponent antibacterial mixture showed powerful antibacterial activities against foodborne pathogens and exhibited better killing effects against *B. cereus* attached to food contact surfaces than chemical antibacterials. The multicomponent antibacterial mixture in this study could provide an alternative, eco-friendly approach for efficient sanitization and reduction of cross-contamination between food contact surfaces and food products.

## Figures and Tables

**Figure 1 foods-09-00059-f001:**
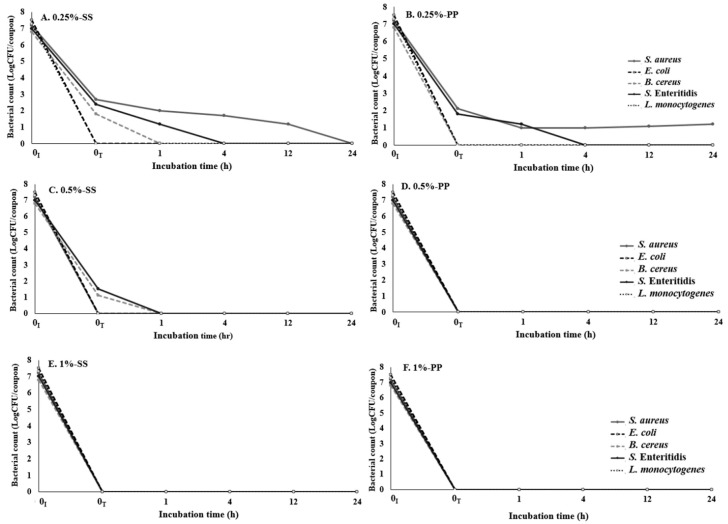
Antibacterial activity of the multicomponent antibacterial mixture against foodborne pathogens on stainless steel (**A**,**C**,**E**) and polypropylene (**B**,**D**,**F**) surfaces. The pathogens were treated with 0.25%, 0.5%, and 1% of the multicomponent antibacterial mixture for 5 min. An average of three repetitions is represented on the graph. Error bars depict the standard error.

**Figure 2 foods-09-00059-f002:**
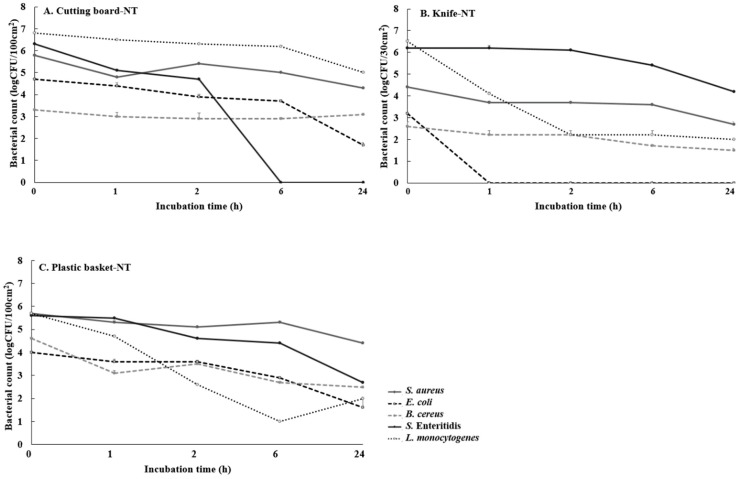
Growth of foodborne pathogens on the cutting board (**A**), knife (**B**), and plastic basket (**C**) surfaces for 24 h without treatment of the multicomponent antibacterial mixture. An average of three repetitions is represented on the graph. Error bars depict standard errors.

**Table 1 foods-09-00059-t001:** Screening of food-grade antibacterial materials (0.5%, *w*/*v*) against foodborne pathogens.

Disinfectant Agent	Inhibition Zone Diameter (mm)				
	Gram-Positive Bacteria		Gram-Negative Bacteria		
	*B. cereus*	*S. aureus*	*L. monocytogenes*	*E. coli*	*S.* Enteritidis
*R. officinalis* L.	23.51 ± 0.08 ^1^	19.42 ± 0.37	14.34 ± 0.42	20.29 ± 0.11	18.97 ± 0.05
*C. sinensis* L.	10.22 ± 0.17	0.0 ± 0.0	0.0 ± 0.0	0.0 ± 0.0	12.44 ± 0.29
Citric acid	9.42 ± 0.27 ^2^	9.04 ± 0.09	0.0 ± 0.0	0.0 ± 0.0	9.44 ± 0.28
ε-polylysine	0.0 ± 0.0	0.0 ± 0.0	0.0 ± 0.0	0.0 ± 0.0	0.0 ± 0.0

^1^ Data are means of two replicates (*n* = 2) ± standard error; ^2^ (-) indicates no growth inhibition zone.

**Table 2 foods-09-00059-t002:** Minimum inhibitory concentration (MIC) values of food-grade antibacterial materials against foodborne pathogens.

Antibacterial Agent	MIC (%, *w*/*v*)				
	Gram-Positive Bacteria			Gram-Negative Bacteria	
	*B. cereus*	*S. aureus*	*L. monocytogenes*	*E. coli*	*S.* Enteritidis
*R. officinalis* L.	0.04 ^1^	0.04	0.04	0.16	0.08
*C. sinensis* L.	0.30	1.25	0.60	1.25	0.30
Citric acid	0.25	0.25	1.0	1.0	0.25
ε-polylysine	>5.0	>5.0	>5.0	>5.0	>5.0

^1^ Data are the means of three replicates (*n* = 3).

**Table 3 foods-09-00059-t003:** MIC values of the multicomponent antibacterial mixture against foodborne pathogens.

Strain	Inhibition Zone Diameter (mm)			
	0.25% ^1^	0.5%	1%	MIC (%, *w*/*v*)
*L. monocytogenes* ATCC 15313	-	9.12	10.41	0.5
*B. cereus* ATCC 14579	8.62	11.04	12.71	0.25
*S. aureus* ATCC 6538	-	-	9.97	1
*E. coli* ATCC 10536	-	11.07	11.34	0.5
*S.* Enteritidis ATCC 13076	-	11.04	12.14	0.5

^1^ 2% formulated antibacterials: *R. officinalis* L. extract, 0.005%; *C. sinensis* L. extract, 0.01%; citric acid, 0.5%; glycerin fatty acid esters, 0.5%; ε-polylysine, 0.01%.

**Table 4 foods-09-00059-t004:** The time (h) required for a 5 log CFU/area reduction in foodborne pathogens on a cutting board, knife, and plastic basket via the use of sodium hypochlorite (200 ppm) and the multicomponent antibacterial mixture (mixture) (0.25%).

Pathogen	T 99.999% (h)					
	Cutting Board		Knife		Plastic Basket	
	NaOCl	Mixture	NaOCl	Mixture	NaOCl	Mixture
*S. aureus*	0_T_ ^1^	0_T_	0_T_	0_T_	0_T_	0_T_
*E. coli*	0_T_	0_T_	0_T_	0_T_	0_T_	0_T_
*B. cereus*	24.0	2.0	6.0	0_T_	6.0	0_T_
*S.* Enteritidis	1.0	0_T_	0_T_	0_T_	0_T_	0_T_
*L. monocytogenes*	0_T_	0_T_	0_T_	0_T_	0_T_	0_T_

^1^ 0_T_ means the initial time after treatment during 5 min.

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
