# Peer review of "The Bactericidal Effect of a Combination of Food-Grade Compounds and their Application as Alternative Antibacterial Agents for Food Contact Surfaces"

_foods, 2020, doi:10.3390/foods9010059_

Round 1

Reviewer 1 Report

The objective of the study is the evaluation of the antimicrobial effect of alternative agents against several microorganisms. The article is interesting and well designed. However, the presentation of the results, data analysis and interpretation of the obtained data need to be improved.

Line 17: Compared to what?

What about the effect of the tested antimicrobial agents on the sensory profile of the food for potential applications, especially taste and odour?

It would be very interesting to discuss the antimicrobial activity of the tested agents based on their active components.

The term “kinetics” cannot be used in the legends of Figures 1 and 2. Kinetic study should include mathematical modelling of the obtained data in order to determine the growth rates of inactivation rate in each case. This should be done in order to make comparisons between the alternative treatments.

English language is poor and should be included significantly throughout the manuscript.

Author Response

Reviewer 1

Line 17: Compared to what?

What about the effect of the tested antimicrobial agents on the sensory profile of the food for potential applications, especially taste and odour?

This study evaluated the application possibility as sanitizer for food contact surfaces such as cutting board, knife, plastic basket and stainless steel and did not consider the potential application for disinfecting foodstuff. However, formulated sanitizer in this study combined with food-grade antimicrobial materials such as rosemary extract, green tea extract, citric acid and polylysine. I think that formulated sanitizer is non-toxic, eco-friendly and hypoallergenic material. Thus, it can commercially use as a disinfectant in a range of foods including dairy, vegetables, meat, fish and fruits for improving food safety. Further study is needed to confirm the potential application for disinfecting foodstuffs throughout considering sensory profiles (taste, odour and browning) of the foods. The contents like this described in Conclusions, Line 404-409.    

It would be very interesting to discuss the antimicrobial activity of the tested agents based on their active components.

Thank you for your comments. I discussed the antimicrobial activity of the tested food-grade antimicrobial materials throughout the investigation based on their active components. The contents inserted in Line 205-219.

The term “kinetics” cannot be used in the legends of Figures 1 and 2. Kinetic study should include mathematical modelling of the obtained data in order to determine the growth rates of inactivation rate in each case. This should be done in order to make comparisons between the alternative treatments.

I accepted your comment and revised the title of Figure 1 and 2.

English language is poor and should be included significantly throughout the manuscript.

The manuscript has been correct any errors in spelling, grammar and word choice throughout English editing service (American Journal Experts, AJE).

Reviewer 2 Report

This manuscript describes a very interesting set of experiments leading to the design of a much-needed food grade disinfectant for food contact surfaces. It is very well written and rather sound from the scientific point of view.

However, one main aspect needs to be improved. The formulation includes a compound (polylysine) as a preservative. The authors failed to acknowledge the possible impact of this compound on the bacteria they are testing as targets for their newly formulated disinfectant. There are literature references mentioning the antibacterial effect of polylysin against, e. g., Bacillus species. The authors should account for the possibility that polylysin is acting as more than a preservative, but also as an active ingredient, that may (at least partially) have accounted for the activity of the combined disinfectant. Further tests should be included in the manuscript to better understand the potential activity of polylysin on the tested bacterial strains - the well-assay test and the MIC determination should be presented also for this ingredient, to improve the scientific soundness of the manuscript.

Other minor comments are given in the attached, annotated manuscript.

Author Response

Major review

The formulation includes a compound (polylysine) as a preservative. The authors failed to acknowledge the possible impact of this compound on the bacteria they are testing as targets for their newly formulated disinfectant. There are literature references mentioning the antibacterial effect of polylysin against, e. g., Bacillus species. The authors should account for the possibility that polylysin is acting as more than a preservative, but also as an active ingredient, that may (at least partially) have accounted for the activity of the combined disinfectant. Further tests should be included in the manuscript to better understand the potential activity of polylysin on the tested bacterial strains - the well-assay test and the MIC determination should be presented also for this ingredient, to improve the scientific soundness of the manuscript.

Thank you for your comments. I agree with your opinion. When I evaluated the antimicrobial activity of food-grade antimicrobial materials, I also tested the antimicrobial activity of polylysine. In 0.5% (w/v) concentration, rosemary extract, green tea extract and citric acid showed anitmicrobial activity against foodborne pathogens while polylysine did not observe antimicrobial activity against foodborne pathogens. Among food-grade antimicrobial materials, rosemary extract, green tea extract and citric acid had MIC values of 0.04-1.25% while polylysine had MIC value of more than 10%. These results did not insert in initial uploaded manuscript. However, I demonstrated that the antimicrobial activity of combined food-grade antimicrobial ingredients observed at lower concentrations than required for the materials alone. The polylysine can change the cell wall composition of bacteria and result in cell wall more fragile, thus increasing the cell wall permeability. The plant extracts also induced cell membrane damage, including having increased outer and inner membrane permeability, disrupted cell membranes. Although alone with polylysine did not show effective antimicrobial activity against foodborne pathogens, the combination of polylysine and plant extracts may be able to damage the bacterial cell membrane through the synergy activity of active compounds, facilitating the absorption of active compounds into cells, thereby increasing the permeability of the cell membrane. The contents like this described in Line 276-283. The results of antimicrobial activity including MIC value of polylysine inserted in Table 1 and 2.

Minor review

Line 37: Please rewrite as: "which is often caused" Not all cases of foodborne intoxications/infections derive from cross-contamination and failures in the hygiene procedures - some are caused by the pre-existing presence of pathogens/their toxins in the raw materials.

->I revised to ‘which is often caused’.

Line 92: What the authors did, in this paper, was designing a compound sanitizer from combinations of individual antimicrobial plant materials. Therefore, I suggest replacing the word "sanitizers" here by "sanitizer ingredients".

I revised to ‘sanitizer ingredients’

Line 97: I presume 20% is the declared concentration of carnosic acid in the ethanolic extract of R. officinalis? If so, please replace "including" with "declared concentration: 20%".

I revised to ‘it contained 20% carnosic acid’

Line 100: Ethanol and methanol are not surfactants...

In this case, they act as solvents. Please replace surfactants with solvents.

I revised to ‘solvents’

Line 101: Please replace "disinfectant agents" with "solutions".

I revised to ‘disinfectant solutions’

Line 104: What was the concentration of poylysine? Being an antimicrobial agent, could it have affected the outcome of the tests?

It has been shown to act synergistically against other Bacillus spp. (see, for instance: Liu et al. (2015) Food Control, 47: 444-450. This might contribute the higher activity of the combined product on B. cereus when compared with hypochlorite. Therefore, polylysine should be regarded as one of the active ingredients in the formula and its concentration should be given.

Thank you for your comments. Polylysine used in this study did not show the antimicrobial activity against foodborne pathogens at 0.5% concentration. The antimicrobial effect of polylysine against tested pathogens showed at a concentration higher than 10% (w/v) and it was very high MIC value compared to other antimicrobial materials. However, like your opinion, combined sanitizer ingredients with polylysine showed effective antimicrobial activity at low concertation compared with individual concentrations against all tested foodborne pathogens on food contact surface. Thus, we inserted the introduction, results and discussion about polylysine as one of the active ingredients in sanitizer formulation. 

Line 104-107: The highlighted sentence belongs to the Results section, not to the Material and Methods.

I inserted the sentence about sanitizer formulation in Results section, Line 251-255.

Line 108: This sentence is not clear. Please reformulate. Enriched 50 fold by comparison with what?

The authors should rather declare what is the concentration of the extracts in the final formulation.

We evaluated the suitable concentration for microbial growth inhibition by time-kill rate and the concentration was 0.005% for R. officinalis L., 0.01% for C. sinensis L., 0.5% for citric acid, 0.01% for polylysine and 0.5% for glycerin fatty acid ester. We made a 50X stock solution to improve the accuracy of concentration. The contents described in Materials and Methods, Line 104-108.  

Line 119: It is costumary to calculate the phenol index in order to compare the efficacy of different antimicrobial agents when using agar diffusion methods. It would have been preferable to do so in this case.

Thank you for your comment. In this study, the antimicrobial activity of food-grade antimicrobial materials was achieved according to the protocols of previous study (Prabuseenivasan, S.; Jayakumar, M.; Ignacimuthu, S. In vitro antibacterial activity of some plant essential oils. BMC Complement Altern Med. 2006, 6, 39-46) and Clinical and Laboratory Standards Institute guidelines (CLSI. 2013. Performance Standards for Antimicrobial Disk Susceptibility Test. Approved Standard (9th edn). Wayne, PA: National Committee for Clinical Laboratory Standards, M2-A9) with some modifications. Agar disk diffusion testing approved by CLSI is the official method used in microbiology laboratories for routine antimicrobial activity testing. We did not consider the phenol index for antimicrobial susceptibility test in this study but our further study will calculate the phenol index in order to compared the efficacy of different antimicrobial agents by agar diffusion method.   

Line 232-233: Please replace "they are" with "they tend to be".

I revised to ‘they tend to be.’

Line 242-243: These data should be added to the relevant part of Materials and Methods (see comments to that section of the manuscript).

The concentration of glycerin fatty acid ester and polylysine was added to Materials and Methods. (Line 107-108)

Line 264: It is also possible that there is an antimicrobial effect of the polylysine itself. This should be duly discussed here, adding the necessary references to substantiate this new part of the discussion.

I accepted your comments and results and discussions about polylysine described in overall manuscript.

Line 333: Again, the authors should account for the possible effect o polylysine on the target bacteria used in this manuscript. Ideally, the antibacterial effect of this component, as well as its MIC against the studied bacteria should have been included in the manuscript.

I accepted your comments and results and discussions about polylysine described in overall manuscript.

Line 366: Please indicate the units (h, min, s?).

I inserted the unit, ‘h’

Line 376: Please take into account what has been previously remarked about polylysine.

The results and discussion about the antimicrobial effects of polylysine included in manuscript.

Round 2

Reviewer 1 Report

I believe that the authors addressed adequately the reviewer's suggestions. However, I cannot see the improvements in the English language in the revised manuscript. Please check again English grammar and spelling and revise accordingly.

Author Response

We checked for correct use of grammar and common technical terms and edited to a level suitable for reporting research in 'Foods' by English editing service of MDPI.  
